# Comparative Na⁺ and K⁺ Profiling Reveals Microbial Community Assembly of Alfalfa Silage in Different Saline-Alkali Soils

**Shengnan Li** [1], **Yushan Bao** [2], **Mingju Lv** [3], **Lianyi Zhang** [3], **Lin Liu** [4], **Yinghao Liu** [5,*] and **Qiang Lu** [1,*]

1   College of Forestry and Prataculture, Ningxia University, Yinchuan 750021, China; l3522293838@163.com
2   Inner Mongolia Autonomous Region Agricultural and Livestock Product Quality and Safety Center, Hohhot 010018, China; baoyushan@sina.com
3   Inner Mongolia Agriculture and Animal Husbandry Extension Center, Hohhot 010000, China; lvmingju@sina.com (M.L.); zhanglianyi@sina.com (L.Z.)
4   Bayannur City Academy of Agricultural Sciences, Bayannur 015000, China; yiyiererwuwu@sina.com
5   Institute of Grassland Research, Chinese Academy of Agricultural Sciences, Hohhot 010000, China
*   Correspondence: liuyinghao@caas.cn (Y.L.); luqiang@nxu.edu.cn (Q.L.)

**Abstract:** Alfalfa cultivated in saline-alkaline soils exhibits a high buffering capacity but low carbohydrate content, posing challenges in the production of high-quality silage feed. This study investigated alfalfa plants grown under varying conditions: mild (QE, salt content 1–2%), moderate (ZE, salt content 2–3%), severe (HE, salt content 3–4%), and non-saline-alkaline (CON, salt content < 1%). Employing a two-factor experimental design, we analyzed the fluctuations in nutritional quality, microbial abundance, and community composition of different salt–alkaline alfalfa materials and silage feeds. Furthermore, we elucidated the fermentation mechanism involved in salt–alkaline alfalfa ensiling. Following a 60-day ensiling period, the ZE and HE treatments led to substantial reductions in pH, acetic acids (AA), branched-chain butyric acids (BA) content, facultative anaerobic bacteria, and *Escherichia coli* populations ($p < 0.05$). Conversely, the ZE and HE treatments increased lactic acid (LA) content and the population of lactic acid bacteria (LAB) ($p < 0.05$). Additionally, these treatments significantly mitigated protein losses in both raw alfalfa and silage feeds ($p < 0.05$), while remarkably augmenting the water-soluble carbohydrates (WSC), Na⁺, and K⁺ content of alfalfa materials. Sodium ions were found to exert a considerable influence on bacterial community composition during salt–alkaline alfalfa ensiling, with *Enterococcus*, *Lactococcus*, and *Lactobacillus* identified as the predominant fermentative microorganisms. Overall, moderately salt-alkaline alfalfa materials displayed optimal nutritional and fermentation quality, ensuring favorable nutritional attributes and fermentation outcomes under such conditions.

**Keywords:** Na⁺; K⁺; alfalfa; microbial community; fermentation characteristics

## 1. Introduction

Soil salinization has emerged as a significant factor constraining global agricultural and livestock development [1]. Approximately 100 million hectares of the world's arable land, out of a total of 1.4 billion hectares, are naturally saline-alkali soils [2]. Furthermore, salt-affected irrigation lands account for approximately 40% of the total irrigated area [3]. The alarming trend is that salt-affected areas worldwide are increasing rapidly at a rate of $1.5 \times 10^7$ hm² per year [4]. Predictions indicate that by 2050, over 50% of arable land will become salt-affected [5]. Soil salinization is now recognized as one of the most prevalent abiotic stresses, posing a threat to global crop and forage yields, as well as their quality [6]. The detrimental effects of high salt content significantly hamper plant growth and development, restricting nutrient absorption and water utilization, thereby leading to reduced crop productivity and compromised food and feed quality. Therefore, addressing

the issue of soil salinization holds paramount importance for global food security and the sustainable development of the livestock industry. However, salt-affected lands also present substantial potential and advantages. By exploring innovative approaches for the development and utilization of salt-affected lands, their unique characteristics can be leveraged to foster advancements in grassland cultivation, livestock management, and solutions to food-related challenges.

Anaerobic fermentation is commonly used in the production of fermented feed by acidifying the samples through the production of organic acids, which inhibits the growth of most aerobic spoilage bacteria. However, the specific salt effects and imbalanced osmotic pressure caused by soil salinization affect the growth and development of most crops, the accumulation of nutrients, and the changes in microbial community structure, subsequently impacting the performance of anaerobic fermentation [7–9]. Our previous studies have indicated that salt stress plays a crucial role in shaping the diverse fermentation processes in alfalfa silage [1]. The types and quantities of plant-associated microorganisms are influenced by various factors, and different plants and the same plant grown in different environments exhibit variations in the types and quantities of surface-associated microorganisms. It has been demonstrated that plant-associated microorganisms respond actively and sensitively to soil salt stress [10]. The differences in the attached microbial community of alfalfa were caused by $Na^+$ content, resulting in changes in silage pH, as well as differential levels of lactic acid and acetic acid. Due to metabolic pathway alterations, the concentration of butyric acid and acetic acid decreased with increasing $Na^+$ concentration, with the decrease in butyric acid concentration being five times greater than that of acetic acid [8]. $Na^+$ affects protease activity during the ensiling process, reducing the production of ammonia nitrogen during ensiling [11]. Adaptation of *Escherichia coli* to high sodium concentrations can enhance cation tolerance and increase lactic acid content [12]. Under salt stress, plants regulate the distribution of $Na^+$ and $K^+$ in their bodies, which not only reduces the physiological toxicity of $Na^+$ but also leads to differences in microbial communities in silage feed. Research has shown that under high $Na^+$ conditions, the growth and reproduction of salt-sensitive bacteria are inhibited, while salt-tolerant bacteria are enriched, which helps to reduce protein degradation and form a stable acidic environment. Additionally, microbial diversity significantly increases with prolonged fermentation periods [9,13].

Silage is a microbial fermentation process that plays a crucial role in determining the quality of silage and the health of ruminant animals. The microbial community structure in silage can vary significantly due to the influence of surface-associated microorganisms on forage and silage conditions. Therefore, it is essential to analyze the dynamic changes in microorganisms during the ensiling process to further improve silage quality. However, to the best of our knowledge, there has been limited research on the mechanism of microbial community dynamics during the ensiling process of salt-tolerant alfalfa in saline-alkali soil. Furthermore, there is limited information on the effectiveness of $Na^+$ and $K^+$ in influencing the fermentation characteristics and microbial community of silage. Therefore, this study aims to explore the impact of different environmental factors on microbial communities and the quality of silage.

Alfalfa, as an excellent leguminous forage, is characterized by high protein content, digestible nutrients, and minerals. It possesses salt and alkali tolerance, drought tolerance, and grazing resistance, making it one of the main cultivated forages and feeds. However, due to the unique characteristics of saline-alkali habitats, the quality and yield of salinized alfalfa hay products are poor, resulting in low conversion efficiency of high-quality forage utilization and severely hampering the development of the salinized forage industry. Therefore, this experiment aims to explore the changes in the nutritional quality of salinized alfalfa under salt stress, elucidate the impact mechanism of $K^+$ and $Na^+$ on the quality and microbial community structure of salinized alfalfa during different stages of ensiling, and provide a theoretical basis for the production of high-quality salinized forage.

## 2. Materials and Methods

### 2.1. Fresh Alfalfa and Ensiling

The study was conducted at the Experimental Station for Forage Processing and Efficient Utilization, Inner Mongolia Agricultural University, Baotou, China ($110°37'$–$110°27'$ E, $40°05'$–$40°17'$ N). The alfalfa used in the experiment was sown in May 2020 using strip seeding method with a row spacing of 15 cm. Soil samples were taken from 10 cm to 20 cm depths at the test site, and the following parameters were measured: pH, total salt contents, $Na^+$, $K^+$, and $Cl^-$ contents (Table 1). Based on the classification standard for saline-alkaline soil, four positions were selected to represent various salt stress levels. The study focused on alfalfa cultivated under mild salinity (QE, salt content 1–2%), moderate salinity (ZE, salt content 2–3%), severe salinity (HE, salt content 3–4%), and non-saline soil conditions (CON, salt content < 1%). Each treatment was replicated three times. The soil physicochemical properties of the experimental site are presented in Table 1. Each experimental plot had an area of 30 m$^2$ (5 m × 6 m).

**Table 1.** The physical and chemical properties of soils in different experimental areas.

| Item | Salt (%) | Na$^+$ (g/kg) | K$^+$ (g/kg) | Cl$^-$ (g/kg) | pH |
|------|----------|---------------|--------------|---------------|-----|
| CON | 0.09 ± 0.01 [c] | 0.11 ± 0.01 [c] | 0.023 ± 0.01 [d] | 0.051 ± 0.02 [c] | 7.44 ± 0.21 [b] |
| QE | 0.20 ± 0.06 [b] | 0.15 ± 0.01 [c] | 0.031 ± 0.01 [c] | 0.119 ± 0.01 [b] | 8.44 ± 0.09 [a] |
| ZE | 0.26 ± 0.05 [b] | 0.16 ± 0.01 [b] | 0.035 ± 0.01 [b] | 0.125 ± 0.01 [b] | 8.61 ± 0.11 [a] |
| HE | 0.48 ± 0.03 [a] | 0.25 ± 0.02 [a] | 0.041 ± 0.01 [a] | 0.203 ± 0.02 [a] | 8.73 ± 0.34 [a] |

[a–d] Means of saline-alkali treatments within a row with different superscripts differ ($p < 0.05$).

Three random samples were selected from each experimental plot at the early flowering stage and cut to a stubble height of 5–8 cm. After harvesting, the alfalfa was immediately transported to the laboratory and air-dried to approximately 50% moisture content. A forage chopper was used to chop the alfalfa into 2–3 cm pieces. Each material was processed separately to prevent cross-contamination. Approximately 600 g of processed alfalfa was packed into a polyethylene bag and vacuum-sealed (DZ-400, Yizhong Machinery Co., Ltd., Zhucheng City, Shandong Province, China) for 60 days at a temperature of 24–26 °C.

### 2.2. Soil Physicochemical Analysis

The soil samples were air-dried indoors, and the soil electrical conductivity (EC) was determined using a conductivity meter (FieldScout EC 110 Meter). The $Na^+$ and $K^+$ ions were measured using atomic absorption spectroscopy [14], while $SO_4^{2-}$ and $Cl^-$ were analyzed using ion chromatography [14]. The soil pH was measured using a PHS-3C acidity meter.

### 2.3. Microbial Enumeration and Silage Quality Analysis

Ten grams of fresh samples or silage feed were mixed with 90 mL of sterile distilled water and shaken at 120 rpm for 2 h. After serially diluting 1 mL of the microbial count solution by a factor of 10, the remaining solution was filtered and stored at −80 °C for DNA extraction. LAB counts were conducted on MRS agar after incubation at 37 °C for 48 h. Enterobacteria counts were performed on VRBG agar after aerobic incubation at 37 °C for 48 h. Aerobic bacteria were cultured on nutrient agar. The viable microbial count per gram of fresh matter was calculated as $\log_{10}$ CFU.

After opening the silage bag, fresh or silage feed samples (20 g) were mixed with 180 mL of distilled water and homogenized using a silage homogenizer for 2 min, followed by filtration through four layers of cheesecloth and filter paper. The pH of the filtrate was measured using a pH meter. Organic acids were measured by high-performance liquid chromatography (KC-811, column, Shodex, Shimadzu, Japan; oven temperature, 50 °C.; flow rate, 1 mL/min; SPD, 210 nm) [15]. Ammonia nitrogen ($NH_3$-N) and water-soluble carbohydrates (WSC) were measured according to Owens [16].

To determine the dry matter (DM) content, alfalfa fresh samples and silage samples were dried at 65 °C for 72 h. The dried samples were ground and passed through a 1 mm sieve for chemical composition analysis. The crude protein (CP) content was determined using the Kjeldahl method [17], while the soluble protein (SP) content was measured following the trichloroacetic acid method [17]. The NDF and ADF contents were determined by the Van Soest fiber analysis method [17]. The concentrations of $Na^+$ and $K^+$ ions in alfalfa were measured using a flame photometer (Model 425) relative to a standard solution.

### 2.4. Bacterial Community of Alfalfa

Microbial DNA from alfalfa silage was extracted using the E.Z.N.A. ® Plant DNA Kit (Omega Bio-Tek, Norcross, GA, USA). The final DNA concentration and purity were determined using a NanoDrop 2000 UV–Vis spectrophotometer (Thermo Scientific, Waltham, MA, USA), and DNA quality was assessed by 1% agarose gel electrophoresis. The V3-V4 region of the 16S rRNA gene was amplified using the thermocycling PCR system (GeneAmp 9700, ABI, Foster city, CA, USA) with primers 338F and 806R. The PCR reactions were conducted using the following program: 3 min of denaturation at 95 °C, 27 cycles of 30 s at 95 °C, 30 s for annealing at 55 °C, and 45 s for elongation at 72 °C, and a final extension at 72 °C for 10 min. PCR reactions were performed in triplicate 20 μL mixture containing 4 μL of 5 × FastPfu Buffer, 2 μL of 2.5 mM dNTPs, 0.8 μL of each primer (5 μM), 0.4 μL of FastPfu Polymerase, and 10 ng of template DNA. PCR products obtained from the 2% agarose gel were further purified using the AxyPrep DNA Gel Extraction Kit (Axygen Biosciences, Union City, CA, USA) and quantified using the QuantiFluor ™-ST kit (Promega, Madison, WI, USA) according to the manufacturer's protocol.

The original fastq files were demultiplexed, quality-filtered using Trimmomatic, and merged using FLASH based on the following criteria: (a) truncating reads at any site receiving an average quality score below 20 in a 50 bp sliding window, (b) removing reads containing ambiguous bases and allowing two-nucleotide mismatches for primer matching, and (c) merging sequences with overlapping lengths greater than 10 bp based on their overlap.

Operational taxonomic units (OTUs) were clustered at a 97% similarity cutoff using UPARSE, and chimeric sequences were identified and removed using UCHIME. Taxonomic classification of each 16S rRNA gene sequence was performed using the RDP classifier algorithm with a confidence threshold of 70% against the Silva (SSU123) 16S rRNA database.

### 2.5. Statistical Analysis

Microbial count data were log-transformed based on fresh weight before statistical analysis. Subsequently, a two-way analysis of variance (ANOVA) based on a 2 × 4 full factorial experimental design (two fermentation time levels × four salt stress levels) was performed to analyze the log-transformed data, chemical composition, fermentation parameters, and α-diversity data. Duncan's test was used for mean comparison between different treatment groups. Statistical significance was considered at $p < 0.05$. The correlation heatmap of the top eighth of the genera and fermentation properties were calculated using the Origin. High-throughput sequencing data were analyzed using the online platform provided by Majorbio I-Sanger Cloud Platform (www.I-Sanger.com accessed on 29 August 2023).

### 3. Results

#### 3.1. Fermentation Characteristics and Chemical Composition of Fresh Triticale and Alfalfa Silage

The fermentation characteristics of alfalfa silage (Table 2) indicated significant effects of salt stress (T), fermentation days (D), and the interaction between salt stress and fermentation days on the content of lactic acid (LA), acetic acid (AA), butyric acid (BA), aerobic bacteria, and *Escherichia coli* ($p < 0.05$). After 60 days of ensiling, the pH values ranged from 4.8 to 5.3. The pH value of the ZE silage was significantly lower than that of other silages, while the LA concentration was significantly higher than that of other silages ($p < 0.05$).

With the rapid decrease in pH, the water-soluble carbohydrate (WSC) content of ZE silage decreased by 146 g/kg compared to before ensiling (248 g/kg vs. 394 g/kg). The AA content of the silages ranged from 32.7 to 47.9 g/kg. As salt stress increased, the accumulation of AA in the four groups of silages showed an increasing and then decreasing trend. The BA content of the silages ranged from 11.4 to 26.9 g/kg. At day 60 of fermentation, the BA content in the CON group was the highest (26.9 g/kg) ($p < 0.05$), while the LA content was the lowest (31.7 g/kg) ($p < 0.05$). After 60 days of ensiling, the counts of aerobic microorganisms in the ZE and HE groups were significantly lower than those in the CON and QE groups, with values of 6.59 and 6.55 $\log_{10}$cfu/g, respectively ($p < 0.05$). Compared to day 0, the counts of general aerobic bacteria in each treatment significantly decreased after 60 days of ensiling ($p < 0.05$), with a reduction range of 4.3% to 24.4%. The count of *Escherichia coli* in the CON group was significantly higher than that in the QE, ZE, and HE groups ($p < 0.05$), while there was no significant difference among the QE, ZE, and HE groups ($p > 0.05$). *Escherichia coli* was not detected after 60 days of ensiling.

**Table 2.** The effects of different treatments on fermentation characteristics of alfalfa.

| Variable | 0 | | | | 60 | | | | SEM | *p*-Value | | |
|---|---|---|---|---|---|---|---|---|---|---|---|---|
| | CON | QE | ZE | HE | CON | QE | ZE | HE | | T | D | T × D |
| pH | 6.42 [b] | 6.20[c] | 6.58 [ab] | 6.63 [a] | 5.19 [ab] | 5.31 [a] | 4.86 [b] | 5.16 [ab] | 0.32 | 0.26 | 0.01 | 0.01 |
| LA (g/kg DM) | 10.77 [a] | 10.53 [a] | 10.23 [a] | 8.43 [b] | 31.70 [c] | 39.47 [b] | 51.03 [a] | 40.23 [b] | 0.32 | <0.01 | <0.01 | <0.01 |
| AA (g/kg DM) | 3.30 [b] | 3.63 [b] | 2.67 [c] | 10.50 [a] | 45.13 [ab] | 47.97 [a] | 32.73 [c] | 42.0 [b] | 0.27 | <0.01 | <0.01 | <0.01 |
| BA (g/kg DM) | 2.45 [b] | 3.13 [a] | 2.07 [c] | 1.13 [d] | 26.93 [a] | 17.20 [b] | 11.40 [c] | 11.53 [c] | 0.19 | <0.01 | <0.01 | <0.01 |
| LAB ($\log_{10}$cfu/g FM) | 6.58 | 6.42 | 6.49 | 6.41 | 7.48 | 7.53 | 7.63 | 7.56 | 0.23 | 0.57 | <0.01 | 0.21 |
| Aerobic bacteria ($\log_{10}$cfu/g FM) | 7.31 [a] | 7.02 [bc] | 6.49 [c] | 7.03 [b] | 6.66 [ab] | 6.73 [a] | 6.59 [b] | 6.55 [b] | 0.12 | <0.01 | <0.01 | <0.01 |
| *Escherichia coli* ($\log_{10}$cfu/g FM) | 4.41 [a] | 4.20 [b] | 4.30 [ab] | 4.18 [b] | ND | ND | ND | ND | 0.13 | <0.05 | <0.01 | <0.01 |

CON, without salt stress; QE, under light salt stress; ZE, under moderate salt stress; HE, under severe salt stress. LA, lactic acid; AA, acetic acid; BA, butyric acid; LAB, lactic acid bacteria. FM, fresh matter. T, salt stress; D, ensiling days; T × D, the interaction between salt stress and ensiling days. SEM, standard error of means. [a–d] Means of inoculation treatments within a row with different superscripts differ ($p < 0.05$).

The chemical composition of alfalfa silage on different days is shown in Table 3. Factor analysis revealed significant effects of T, D, and the interaction between T and D on the content of NH$_3$-N, soluble protein (SP), and acid detergent fiber (ADF) ($p < 0.01$). Compared to the alfalfa materials in each group, the SP content increased by 35–81 g/kg. After 60 days of fermentation, the SP content in the QE group (153 g/kg) was significantly higher than that in the other groups. Meanwhile, the NH$_3$-N content was lower, which might inhibit further degradation of crude protein (CP) by proteases and eventually lead to the accumulation of SP. Salt stress had a significant impact on CP, ADF, and Na$^+$ content ($p < 0.01$). As the fermentation time prolonged, the ADF content of alfalfa decreased ($p < 0.05$). The ADF content in the HE group (35.20%) was significantly higher than that in the CON (32.30%), QE (32.33%), and ZE (34.33%) groups ($p < 0.05$). After 60 days of salt stress, the Na$^+$ content in the HE group was significantly higher than that in the CON, QE, and ZE groups ($p < 0.01$). The interaction between salt stress and fermentation days had a significant effect on the K$^+$ content ($p < 0.05$).

### 3.2. Microbial Composition of Alfalfa Silage

Alpha diversity refers to the diversity within a specific region or ecosystem, which can reflect the abundance and diversity of microbial communities (Table 4). After 60 days of ensiling, the Sobs, Shannon, and Chao1 indices of all treatment groups were lower than those of fresh alfalfa, indicating a decrease in bacterial community diversity, species richness, and abundance. The coverage index for each treatment group was greater than 99%. Principal coordinate analysis (PCoA) based on Bray–Curtis distances was used to identify variations in the bacterial communities between fresh alfalfa samples and fermented silage under different levels of salt stress (Figure 1a). The first two principal coordinates (PC1 and PC2)

accounted for 60.98% and 14.14% of the total variance, respectively. Bacterial communities were divided into two clusters, one representing the bacterial community structure of raw alfalfa from saline-alkali soil, and the other representing the bacterial community structure of alfalfa silage after 60 days of fermentation.

**Table 3.** The effects of different treatments on the nutrient quality content of alfalfa.

| Variable | 0 | | | | 60 | | | | SEM | *p*-Value | | |
|---|---|---|---|---|---|---|---|---|---|---|---|---|
| | CON | QE | ZE | HE | CON | QE | ZE | HE | | T | D | T × D |
| DM (g/kg FM) | 304.33 | 312.67 | 319 | 305.33 | 270.33 | 267.67 | 269.33 | 274.33 | 1.57 | 0.51 | < 0.01 | 0.15 |
| CP (g/kg DM) | 192.33 [c] | 207.83 [b] | 234.30 [a] | 207 [b] | 172.67 [c] | 183.33 [bc] | 215 [a] | 189.33 [b] | 1.33 | < 0.01 | < 0.01 | 0.82 |
| NH$_3$-N total N | 8.53 [a] | 5.77 [c] | 7.20 [b] | 4.53 [d] | 33.57 [a] | 29.46 [b] | 18.40 [c] | 22.60 [c] | 0.46 | < 0.01 | < 0.01 | < 0.01 |
| SP (g/kg DM) | 63.33 [c] | 72.33 [b] | 103.67 [a] | 68.33 [bc] | 144.67 [ab] | 153 [a] | 138.33 [b] | 141.67 [b] | 0.90 | < 0.01 | < 0.01 | < 0.01 |
| NDF (g/kg DM) | 432 [b] | 420.33 [c] | 474.33 [a] | 467.67 [a] | 403.33 [b] | 388 [c] | 438.33 [a] | 433.67 [a] | 0.78 | < 0.01 | < 0.01 | < 0.01 |
| ADF (g/kg DM) | 353.67 [c] | 348 [d] | 386.33 [a] | 381 [b] | 323 [c] | 323.33 [c] | 343.33 [b] | 352 [a] | 0.55 | < 0.01 | < 0.01 | < 0.01 |
| WSC (g/kg DM) | 366.33 [c] | 372.33 [b] | 394 [a] | 368.33 [bc] | 304.67 | 306.33 | 248.33 | 305 | 6.98 | 0.79 | < 0.01 | 0.13 |
| Na$^+$ (g/kg DM) | 1.98 [d] | 2.74 [c] | 4.83 [b] | 5.10 [a] | 2.22 [b] | 2.60 [b] | 4.56 [a] | 4.72 [a] | 0.05 | < 0.01 | 0.19 | 0.21 |
| K$^+$ (g/kg DM) | 26.73 [d] | 31.23 [a] | 30.93 [b] | 29.43 [c] | 29.9 | 29.2 | 29.9 | 29.67 | 0.26 | 0.05 | 0.87 | < 0.05 |

CON, without salt stress; QE, under light salt stress; ZE, under moderate salt stress; HE, under severe salt stress. DM, dry matter; CP, crude protein; NH$_3$-N, ammonia nitrogen; SP, soluble protein; NDF, neutral detergent fiber; ADF, acid detergent fiber; WSC, water-soluble carbohydrate; FM, fresh matter. T, salt stress; D, ensiling days; T × D, the interaction between salt stress and ensiling days. SEM, standard error of means. [a–d] Means of inoculation treatments within a row with different superscripts differ (*p* < 0.05).

**Table 4.** Analysis of alfalfa silage's alpha diversity.

| Variable | 0 | | | | 60 | | | | SEM | *p*-Value | | |
|---|---|---|---|---|---|---|---|---|---|---|---|---|
| | CON | QE | ZE | HE | CON | QE | ZE | HE | | T | D | T × D |
| Sobs | 71.00 | 70.67 | 60.67 | 62.33 | 56.33 | 66.67 | 58.67 | 43.00 | 2.11 | 0.09 | <0.05 | 0.43 |
| Shannon | 1.92 | 1.87 | 1.69 | 1.84 | 1.43 | 1.61 | 1.51 | 1.15 | 0.07 | 0.65 | <0.05 | 0.57 |
| Ace | 81.16 | 85.14 | 77.48 | 75.16 | 87.61 | 76.42 | 88.97 | 98.00 | 4.47 | 0.97 | 0.38 | 0.66 |
| Chao1 | 77.45 | 81.24 | 86.53 | 72.66 | 70.00 | 75.33 | 80.22 | 69.89 | 3.12 | 0.55 | 0.38 | 0.99 |
| Coverage | 0.9989 | 0.9990 | 0.9987 | 0.9989 | 0.9991 | 0.9992 | 0.9986 | 0.9987 | 0.00 | 0.31 | 0.78 | 0.82 |

CON, without salt stress; QE, under light salt stress; ZE, under moderate salt stress; HE, under severe salt stress. T, salt stress; D, ensiling days; T × D, the interaction between salt stress and ensiling days. SEM, standard error of means.

To evaluate the bacterial community at the genus level under salt stress and different treatments, we used the Kruskal–Wallis H test (Figure 1b). After 60 days of fermentation, the abundance of *Pantoea*, *Pseudomonas*, *Methylobacterium*, and *Rhizobium* decreased, while *Lactococcus*, *Lactobacillus*, *Escherichia-Shigella*, and *Enterococcus* increased. Specifically, the abundance of *Pantoea* decreased with increasing salt stress. The abundance of *Lactococcus* in salt-stressed silage was lower than that in CON-60, with the highest abundance observed in the HE-60 treatment group. Compared to other treatment groups, the ZE-60 treatment group had the highest abundance of *Lactobacillus* and the lowest abundance of *Escherichia-Shigella*. The abundance of *Enterococcus* in the ZE-60 and HE-60 treatment groups was significantly lower than that in CON-60, while the QE-60 treatment group had a higher abundance than CON-60. The abundance of *Rhizobium* decreased with increasing salt stress. *Sphingomonas* and *Methylobacterium* exhibited an increasing trend with increasing salt stress, followed by a decrease, and showed the same pattern after 60 days of ensiling.

To further understand the dynamics of bacterial community succession in alfalfa silage under salt stress, bacterial communities at the phylum and genus levels were assessed (Figure 2). The bacterial communities of fresh alfalfa and ensiled feed were primarily composed of four phyla (Figure 2a). Before ensiling, *Proteobacteria* had the highest abundance, followed by *Actinobacteria*, *Bacteroidetes*, and *Firmicutes*. After ensiling, *Firmicutes* became the dominant phylum. Compared to other treatment groups, *Firmicutes* had a higher abundance in the ZE-60-treated silage. At the genus level, the predominant genera in the pre-ensiling treatments were *Pantoea*, *Pseudomonas*, and *Sphingomonas*. In the ZE treatment, *Chryseobacterium* and *Pseudomonas* had higher relative abundances compared to other treatment groups, while *Pantoea* had a lower relative abundance. After 60 days

of ensiling, the dominant genera in all treatment groups were *Lactococcus*, *Lactobacillus*, *Escherichia-Shigella*, and *Enterococcus* (Figure 2b). *Escherichia-Shigella* and *Enterococcus* were dominant in the CON-60 and QE-60 treated silage. *Lactococcus* and *Lactobacillus* became dominant in the ZE-60 and HE-60 treatments, respectively. The lowest concentrations of *Escherichia-Shigella* and *Enterococcus* were observed in the ZE-60-treated silage.

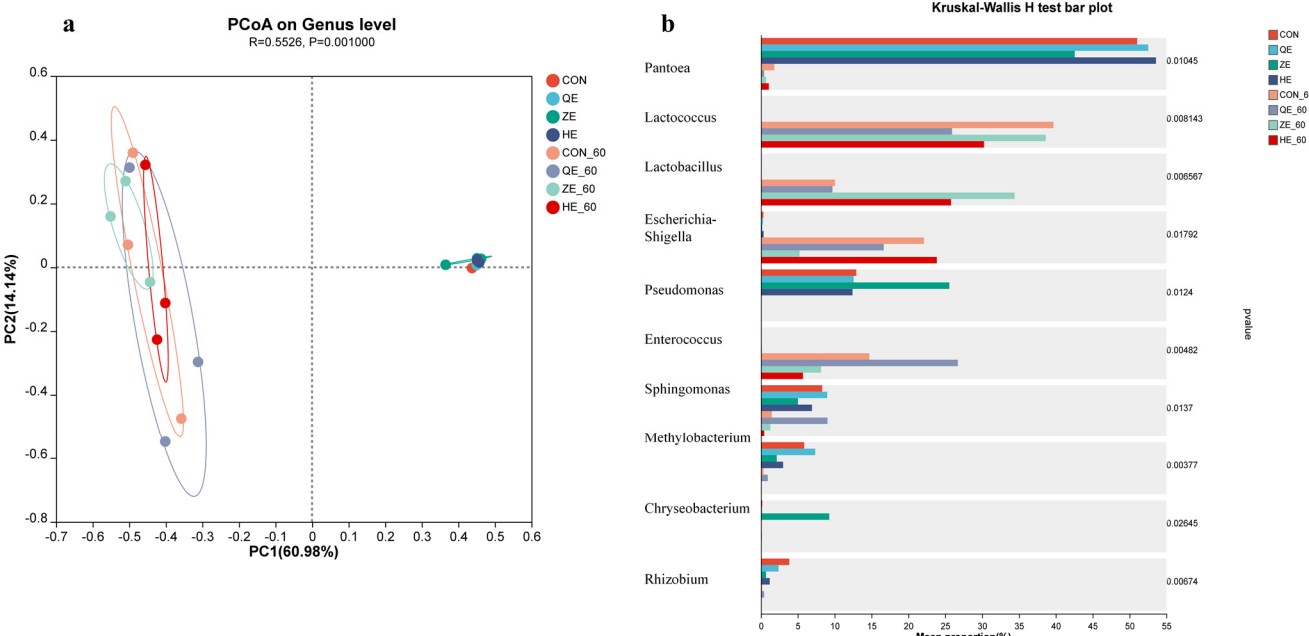

**Figure 1.** Bacteria community principal coordinates analysis (PCoA) on genus level. (**a**) Comparison of microbial variations using the Kruskal–Wallis H test (**b**) for raw alfalfa and alfalfa silage. CON, fresh material of alfalfa without salt stress; QE, alfalfa under light salt stress; ZE, alfalfa under moderate salt stress; HE, alfalfa under severe salt stress; 60, 60 days of ensiling.

Circos plots illustrated the dynamic changes in bacterial community composition at the genus level during alfalfa ensiling (Figure 2c). As the fermentation time increased, the relative abundance of *Pantoea* decreased, while *Lactococcus* and *Lactobacillus* gradually became dominant. In the QE-60 ensiling, *Enterococcus* continued to proliferate after suitable anaerobic conditions were established, inhibiting the growth of LAB, resulting in a gradual decrease in the abundance of *Lactococcus* and *Lactobacillus* and an increase in *Enterococcus* abundance. Additionally, excessively low or high levels of salt stress were unfavorable for suppressing the growth of *Escherichia-Shigella*.

The main nutrients and microbial first-factor and second-factor loadings during the ensiling process of alfalfa on different saline-alkali soils are shown in Figure 3. During the ensiling process, a negative correlation was observed between general aerobic bacteria, molds, *Escherichia coli*, and CP, WSC, ADF, and NDF in the CON-60, QE-60, ZE-60, and HE-60 treatments. In contrast, a positive correlation was observed between LAB and CP, WSC, ADF, and NDF. In the ZE-60 treatment, a positive correlation was observed between general aerobic bacteria, *Escherichia coli*, CP, WSC, ADF, and NDF, while a negative correlation was found with molds. Based on the correlation analysis between the main nutrients and microorganisms during the ensiling process of alfalfa on saline-alkali soils, a higher quantity of LAB indicates better nutritional quality, while higher quantities of *Escherichia coli*, molds, and general aerobic bacteria are detrimental to the nutrient preservation of *Medicago sativa* in the ensiling process. In the ZE-60 treatment, molds showed a negative correlation with the main nutrients. From the above analysis, it can be concluded that LAB are the key factors influencing the nutritional quality of *Medicago sativa* during the ensiling process in different saline-alkali soil conditions.

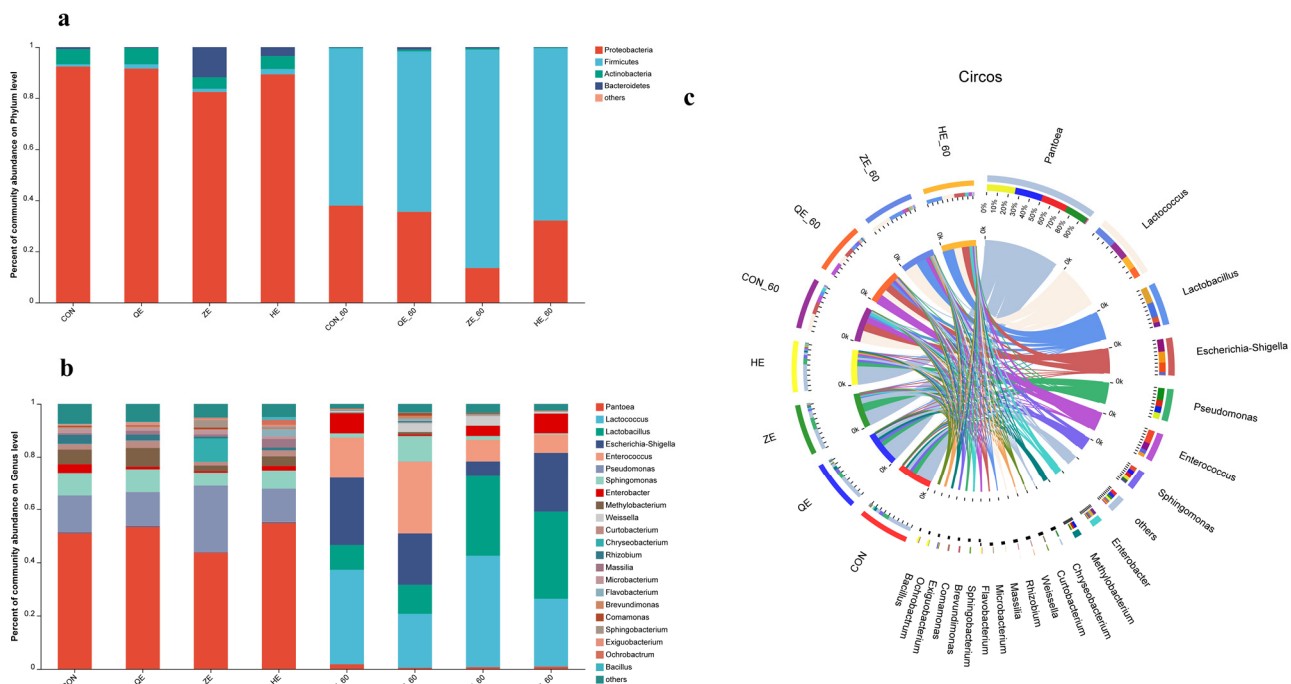

**Figure 2.** Bacterial communities and relative abundance by phylum level (**a**,**c**) and genus level (**b**) for raw alfalfa and alfalfa silage. CON, fresh material of alfalfa without salt stress; QE, alfalfa under light salt stress; ZE, alfalfa under moderate salt stress; HE, alfalfa under severe salt stress; 60, 60 days of ensiling.

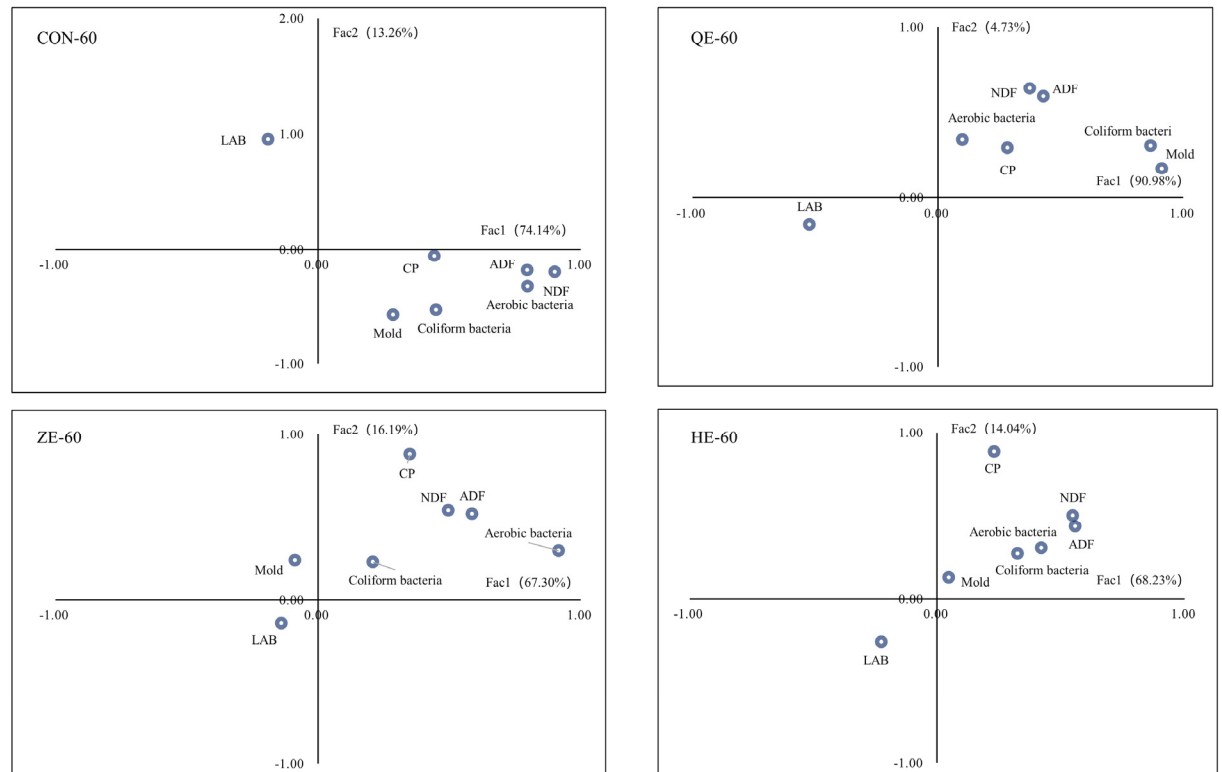

**Figure 3.** The nutrient index and fermentation factor loading diagram of silage under different treatments.

The first and second principal component loadings for different alfalfa ensiling processes in different salt–alkali soils are shown in Figure 3. During ensiling, CON-60, QE-60,

ZE-60, and HE-60 were closely related to CP, WSC, ADF, and NDF, showing a positive correlation. BA, AA, LA, and $NH_3$-N were negatively correlated with CP, WSC, ADF, and NDF, indicating that pH is a key factor impacting the nutrition in different salt–alkali soil alfalfa ensiling.

### 3.3. Correlation between the Microorganism and Fermentation Parameters

pH and WSC were significantly negatively correlated with *Enterococcus* ($p < 0.01$) (Figure 4). AA showed a significant positive correlation with *Lactobacillus* ($p < 0.01$) and a highly significant positive correlation with *Lactococcus* ($p < 0.001$). CP was significantly negatively correlated with *Enterobacter* ($p < 0.01$). SP showed a significant positive correlation with *Lactococcus* and *Lactobacillus* ($p < 0.01$), but a highly significant negative correlation with *Pseudomonas* ($p < 0.001$). $NH_3$-N was significantly negatively correlated with *Escherichia-Shigella* ($p < 0.01$) and highly negatively correlated with *Pseudomonas* ($p < 0.001$). ADF showed a significant negative correlation with *Lactococcus* and *Escherichia-Shigella* ($p < 0.01$) and a significant positive correlation with *Pseudomonas* ($p < 0.01$).

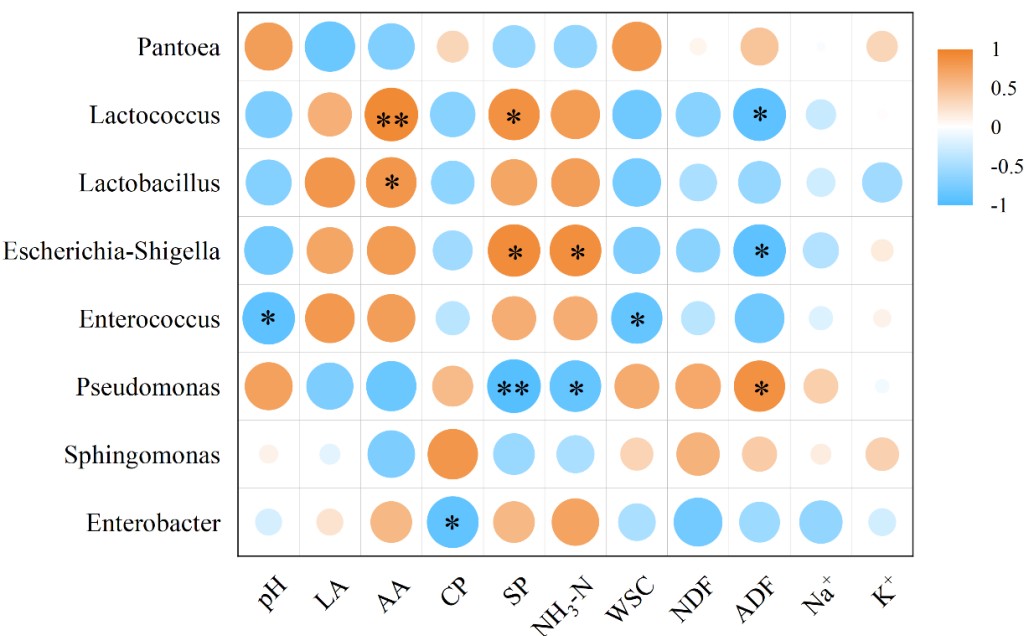

**Figure 4.** The correlation between microorganisms and fermentation parameters at the genus level using Spearman's correlation analysis. * $p < 0.01$; ** $p < 0.001$.

## 4. Discussion

The research findings indicate that after 60 days of ensiling, the pH decreased and the organic acid content increased in all treatment groups, which was closely related to the increase in the number of LABs such as *Lactobacillus* and *Lactococcus*. The formation of an anaerobic environment and the rapid accumulation of lactic acid inhibited the growth and reproduction of aerobic bacteria such as aerobic bacteria and *Escherichia coli*. It is also possible that sodium ions inhibited the growth of *Escherichia coli* through oxidative damage [18]. The pH value of silage is an important evaluation index for assessing the fermentation effect. The pH value of well-fermented silage should be 4.2 or lower [19]. In this study, the pH of all the silages was higher than 4.2, indicating poor fermentation quality. The variation in pH may be due to differences in the adherent microbial population of alfalfa grown in different saline-alkali soils. Salinity, as a measure of salt concentration in soil or water, can have a notable influence on pH levels [20]. The presence of high salt concentrations, particularly sodium ions, can lead to alkaline conditions in affected environments. This relationship arises from the fact that sodium ions can displace other cations in soil, such as calcium, magnesium, and potassium, leading to an increase in soil

pH. Moreover, the presence of excess salts can adversely affect plant root function and impede nutrient uptake, resulting in restricted plant growth and reduced organic matter decomposition. These effects can indirectly impact soil pH by altering microbial activity and nutrient cycling processes. Conversely, changes in pH levels can also influence salinity dynamics. pH alterations can affect the solubility and availability of salts. For example, under acidic conditions, certain salts may dissociate and release their constituent ions, thereby increasing the salt concentration and subsequently the salinity of the environment. The differences in the chemical composition of alfalfa raw materials may be the reason for the significant differences in the fermentation characteristics of silage, which was also confirmed in the study of silage feed of *Trifolium angustifolium* by Xue et al. [21]. The lower pH and higher lactic acid content in ZE-60 and HE-60 were due to the rapid conversion of water-soluble carbohydrates (WSC) into lactic acid (LA) by LAB, resulting in a decrease in pH and stability of silage in a short period of time [22]. Acetic acid (AA) is derived from the decomposition and fermentation of sugars by heterofermentative LAB, *Enterobacteriaceae*, and *Clostridium* [22]. Butyric acid (BA) and other organic acids reflect the efficiency of fermentation or secondary fermentation [23]. In addition, the content of BA also depends on the content of LA, which may be caused by the secondary fermentation of heterofermentative LAB and yeast [24].

Soluble carbohydrates and starch are the main forms of energy storage in plants. With the increase in salt concentration, the content of WSC showed an increasing trend followed by a decreasing trend. When the soil salinity was 2–3%, the content of WSC in alfalfa was the highest. The changes in crude protein (CP) and WSC were consistent, which is similar to the research results of Morais et al., indicating that salt stress as a driving factor alters the nutrient content in alfalfa and appropriate salt stress promotes the accumulation of CP and WSC [25]. Carbohydrates, in the form of sugars and sugar alcohols, play a crucial role as osmoprotectants in plants under osmotic stress [26]. Osmotic stress occurs when plants are exposed to elevated salt levels in their environment. Osmotic stress disrupts the osmotic balance within the plant cells, leading to water loss and dehydration. To counteract this water loss, plants respond by accumulating compatible solutes or osmoprotectants, such as carbohydrates, which help maintain cellular osmotic potential and restore cellular hydration. Salt stress inhibits the growth–nutrient accumulation stage of alfalfa, while the synthesis of small molecular substances in plants can help regulate their own osmotic pressure and enhance their ability to adapt to salt stress. In this study, salt-tolerant alfalfa grown in saline-alkali soils accumulated soluble protein (SP) to counteract salt stress. Soluble carbohydrates in silage feed are the fermentation substrates necessary for maintaining normal microbial metabolic activities [27]. In this study, soluble carbohydrates were metabolized and utilized by LAB to produce organic acids. However, *Weissella* and *Enterobacter* found in the 60-day silage not only consumed a large amount of WSC but also reduced the utilization rate of WSC [28]. The degradation of structural carbohydrates during ensiling increased the content of soluble carbohydrates in silage and provided energy substances for the metabolic fermentation of LAB. Compared with the raw material, the neutral detergent fiber (NDF), and acid detergent fiber (ADF) contents in silages of different saline-alkali soils decreased, thereby supplementing the fermentation substrates of LAB and reducing losses. The accumulation of ammonia nitrogen ($NH_3$-N) is mainly driven by plant proteases and protein hydrolytic enzymes [11], and the $Na^+$ content has an impact on the activity of proteases [29]. When the $Na^+$ concentration is below 60 mg/L, $Na^+$ promotes the activity of proteases, while when the $Na^+$ concentration is above 60 mg/L, $Na^+$ inhibits the activity of proteases. In this study, the $Na^+$ in various treatment groups inhibited the activity of proteases.

The presence of $NH_3$-N in silage is an important indicator reflecting the degree of protein hydrolysis. Ammonia nitrogen accumulation is typically caused by protein hydrolytic enzymes [30]. Even though most plant proteases in alfalfa silage show higher activity at pH 5.0–6.0 [31], excessive ions can inhibit the activity of proteases [29]. This may be the reason why $Na^+$ inhibits the activity of proteases during ensiling. The inhibitory effect of ZE-60

and HE-60 on ammonia nitrogen accumulation suggests that a higher sodium ion content promotes protein preservation during ensiling. After fermentation, large molecule proteins in alfalfa were broken down into small molecule proteins with WSC [32], and the content of SP increased from 63–103 g/kg to 138–153 g/kg. The analysis of nutritional quality and fermentation quality factors of different salt-tolerant alfalfa silages confirms that pH is a key factor affecting nutrition, particularly the correlation between alfalfa CP and pH in the QE-60 treatment group.

Compared to the raw material, the anaerobic environment caused changes in the microbial habitat, suppressing aerobic microorganisms during the ensiling process, and resulting in differences between fresh lucerne and silage microbial communities [22]. The coverage of all samples exceeded 99%, indicating comprehensive sequencing and sufficient data representation of the bacterial community.

In this study, *Proteobacteria*, *Firmicutes*, *Actinobacteria*, and *Bacteroidetes* were dominant phyla in all lucerne samples. These findings are consistent with previous studies on lucerne and corn silage [24,28]. *Firmicutes* and *Proteobacteria* were the most abundant phyla in the silage fermentation process [33]. Nazar et al. reported a transition in the bacterial community from *Proteobacteria* to *Firmicutes*, revealing the significant role of anaerobic and acidic conditions in maintaining the growth of *Firmicutes* [22]. In addition, the relative abundance of *Proteobacteria* exhibited a regular decrease followed by an increase as the salt stress level increased. Luttenton et al. pointed out that the epiphytic plant communities vary under different environmental conditions [34]. Bacteroidetes constituted a significant proportion of bacteria in the ZE and HE treatment groups, with higher relative abundance observed in the ZE treatment group. This suggests that *Bacteroidetes* may be better adapted to environments with higher salt content; however, when the bacteria reach a tolerance threshold, salt ions in plants can inhibit their growth and reproduction. The response of plant-associated microbes to different salt stresses is intriguing and beyond expectation. With increasing salt stress, *Pantoea*, *Pseudomonas*, *Sphingomonas*, and *Methylobacterium* exhibited distinct response patterns. Interestingly, the relative abundance of *Pseudomonas* increased as the relative abundance of *Pantoea* decreased. It has been reported that *Pantoea* and *Pseudomonas*, as plant growth-promoting rhizobacteria (PGPR), can produce IAA and promote biomass under salt stress conditions [35], helping plants survive better. However, *Pseudomonas* exhibits a stronger adaptability to saline environments than *Pantoea* [36], resulting in an increase in its abundance. *Sphingomonas* and *Methylobacterium* have limited adaptation to saline environments, with their relative abundances increasing first and then decreasing with increasing salt stress. Moreover, *Methylobacterium*, a genus within *Proteobacteria*, possesses excellent colonization characteristics in salt-stress environments [37], and the produced IAA and cytokinins contribute to better plant growth. Additionally, the presence of *Methylobacterium* can alleviate the impact of salt stress on plants.

After the fermentation process, *Lactococcus*, *Lactobacillus*, *Escherichia-shigella*, and *Enterococcus* were dominant genera. The lower pH inhibited the growth and reproduction of *Pantoea*, *Pseudomonas*, and *Methylobacterium*. Under anaerobic conditions, *Enterococcus* became the dominant genus in the CON-60 and QE-60 treatment groups. Wang et al. reported that *Enterococcus* is commonly used to enhance fermentation characteristics [23]. *Enterococcus* can rapidly produce LA in the early stage of fermentation, establishing an acidic anaerobic environment conducive to the growth of LAB. However, *Enterococcus* is not an acid-resistant genus [38], which is consistent with the findings of this study. In the QE-60 treatment group, *Sphingomonas* had a relative abundance value of 16.1%, which was higher than that in other treatment groups. *Sphingomonas* can lead to further hydrolysis of CP and SP in silage, resulting in an increase in $NH_3$-N content [39].

*Lactococcus* is commonly found in naturally fermented silage. This contradicts the findings of Wang et al. [23], which may be attributed to the fact that this study did not use any additives and relied solely on the microbes attached to lucerne under salt stress. *Lactobacillus* and *Lactococcus* play important roles in pH reduction and LA accumulation and are widely present throughout the ensiling process. *Lactobacillus* has a stronger acid

tolerance than cocci LAB and can therefore grow vigorously in the late fermentation stage. *Lactobacillus* became the dominant genus in the ZE-60 and HE-60 treatment groups due to its stronger acid tolerance than *Lactococcus*. Generally, *Lactobacillus* is the main producer of LA in silage, while *Enterococcus* cannot survive at lower pH due to its weaker acid resistance. In this study, after 60 days of ensiling fermentation of salt–alkali tolerant alfalfa under salt stress conditions, the content of CP and WSC decreased to varying degrees. This is because microorganisms such as *Sphingomonas*, *Enterobacter*, and *Weissella* consume a large amount of nutrients during their metabolic processes. These microorganisms not only deplete soluble carbohydrates but also have a low utilization rate of soluble carbohydrates.

The findings of this study demonstrate a significant correlation between the chemical composition of silage feed and the microbial community. Variations in the chemical composition and microbial communities of alfalfa under different salt stress conditions contribute to the distinct characteristics of silage microbial communities. This phenomenon may be attributed to the microbial utilization of organic compounds such as starch and organic acids for energy production, as they serve as a source of chemotrophic organic nutrients. Certain microbes, such as *Pseudomonas*, *Bacteroides*, and *Stenotrophomonas*, consume proteins, while others, including *Megamonas*, *Raoultella*, *Citrobacter*, and *Enterococcus*, ferment carbohydrates [28,40]. These results further validate the significant relationship between the chemical composition of the raw materials and silage feed.

Interestingly, $Na^+$ and $K^+$ inhibit the growth and reproduction of most microorganisms. Although the ions themselves do not directly participate in microbial growth, bacteria can achieve intercellular communication through potassium ion channels' electrical signals [7]. This explains the lower microbial abundance and diversity observed under salt stress conditions. Ion stress not only suppresses microbial activity and abundance but also inhibits the proliferation of harmful microorganisms. Therefore, a clear understanding of microbial composition and distribution is crucial for ecological and biogeographical studies, as well as for the conservation and utilization of microbial resources.

Moreover, the inhibitory effect of sodium and potassium ions on microbial growth and reproduction has significant implications for agricultural practices and livestock nutrition. The presence of excessive salt in silage feed can lead to suboptimal microbial activity, resulting in decreased feed quality and digestibility. This, in turn, can negatively impact livestock health and performance.

To address these challenges, it becomes crucial to explore strategies that mitigate the adverse effects of salt stress on microbial communities in silage. One potential approach is to optimize the chemical composition of silage feed through proper selection and management of raw materials. By carefully balancing the levels of sodium and potassium ions, it may be possible to create an environment that promotes the growth of beneficial microorganisms while suppressing the proliferation of harmful ones.

Furthermore, understanding the dynamics of microbial composition and distribution under salt stress conditions can provide valuable insights for the development of targeted interventions. For instance, probiotics or microbial additives could be incorporated into silage feed formulations to enhance the resilience and metabolic activity of desirable microorganisms. These interventions have the potential to improve silage fermentation, increase nutrient availability, and ultimately enhance animal productivity and health.

## 5. Conclusions

This study examined the microbial composition and ensiling quality of salt-stressed alfalfa. The ensiling quality of alfalfa in medium saline soil was the best. Furthermore, under salt stress conditions, beneficial genera such as *Lactococcus* and *Lactobacillus* dominate the microbial community in alfalfa silage, leading to improved fermentation quality. These findings emphasize that salt stress and plant ions play a crucial role in shaping the diverse fermentation processes in alfalfa silage. The identification of an exogenous symbiotic microbial community in salt-stressed alfalfa offers promising prospects as a valuable biological resource for enhancing fermentation quality. It is worth noting that the ecological insights

derived from this research extend beyond agriculture, contributing to our understanding of microbial ecology and resource management in saline environments.

**Author Contributions:** Conceptualization, Q.L., Y.L. and Y.B.; methodology, Q.L.; formal analysis, L.L., Y.B. and M.L.; resources, Q.L and Y.L.; data curation, Y.L, Y.B. and M.L.; writing—original draft preparation, S.L., M.L., L.L. and L.Z.; writing—review and editing, Q.L., S.L. and L.Z.; super-vision, Q.L. and S.L. All authors have read and agreed to the published version of the manuscript.

**Funding:** This work was supported by the Ningxia Higher Education Institutions First-Class Discipline Construction Project (NXYLXK2017A01), the Fundamental Research Funds of the Chinese Academy of Agriculture (110233160007007), Inner Mongolia Autonomous Region Science and Technology Planning Project (2022YFHH0046), Inner Mongolia Natural Science Foundation Project (2022LHQN3003).

**Institutional Review Board Statement:** Not applicable.

**Informed Consent Statement:** Not applicable.

**Data Availability Statement:** Not applicable.

**Acknowledgments:** The authors are grateful for the support provided by the State Power Shengli Energy Co., Ltd. Shared Services Center.

**Conflicts of Interest:** The authors declare no conflict of interest.

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
