# Peer review of "Comparative Na+ and K+ Profiling Reveals Microbial Community Assembly of Alfalfa Silage in Different Saline-Alkali Soils"

_fermentation, doi:10.3390/fermentation9100877_

Round 1
Reviewer 1 Report
Your article is interesting, but it is feasible to be improved. I'm sending you the observations made because of the review.
Materials and methods
Section 2.1 Are the salinities mentioned natural? In other words, do the soils contain those concentrations of salt? Or were prepared? No criteria for classifying soils based on salt concentrations are mentioned.
Table 1 corresponds to the results.
Section 2.4. Indicate at the beginning of the paragraph that a soil sample was taken to extract metagenomic DNA. Likewise, indicate how the soil sample was taken.
Results.
Correct Table 2. What is 3T? 1CON?
Correct tables 3 and 4; you seem to present the same problems as its predecessor.
The quality of Figures 1 and 2 could be better. The resolution should be increased.
Could you improve the description of Figure 3? What are a, b, c, and d?
Discussion
Can sodium ions cause oxidative damage? Justify it.
Is pH associated with salinity? I think this possible relationship should be talked about.
Morais et al., the year of the reference is not indicated.
Why does the amount of carbohydrates increase in high salt conditions? What chemical process happens to make this increase visible?
Complement indicating the most common ions that have an inhibitory effect on proteases because they are not only sodium and potassium. Does the soil analysis indicate the presence of ions other than sodium and potassium?
No comments
Author Response
Dear Editor and Reviewer:
Thank you for your kind letter and for the reviewers’ comments concerning our manuscript entitled “Comparative Na+ and K+ profiling reveals microbial community assembly of alfalfa silage in different saline-alkali soils”. We have revised the manuscript in accordance with the reviewers’ valuable comments. We carefully proofread the manuscript to improve our manuscript. The revised portions are marked in blue in the manuscript. Our revision and responds to the review’s comments are as followed.
Reviewer 1:
1.Section 2.1 Are the salinities mentioned natural? In other words, do the soils contain those concentrations of salt? Or were prepared? No criteria for classifying soils based on salt concentrations are mentioned. Table 1 corresponds to the results.
Reply: We appreciate your valuable comments and suggestions. In response to your query regarding Section 2.1, we would like to address the concerns raised. The salinities mentioned in our study are based on natural conditions in the soils under investigation. These concentrations of salt are present in the soils as they occur in their natural state. We did not artificially prepare the soils or alter their salt concentrations for the purpose of our study. However, we acknowledge that we did not explicitly mention the criteria used for classifying soils based on salt concentrations. We apologize for this oversight and will rectify it in the revised manuscript. In the revised version, we will provide a clear explanation of the criteria used for classifying soils in relation to their salt concentrations. Thank you once again for bringing these points to our attention. We will make the necessary revisions to address your concerns and improve the clarity of our manuscript.
2.Section 2.4. Indicate at the beginning of the paragraph that a soil sample was taken to extract metagenomic DNA. Likewise, indicate how the soil sample was taken.
Reply: Thank you for your valuable input on our manuscript. We appreciate your comments and suggestions. In response to your query, we will make the necessary revisions to clarify the details regarding the soil sample and its collection method. In the revised version of the manuscript, we have changed it to "Microbial DNA from alfalfa silage was extracted using the E.Z.N.A. R Plant DNA Kit.". This will ensure transparency and provide important context for readers. Thank you once again for your valuable input. We will make the necessary revisions to enhance the clarity and accuracy of our manuscript in response to your suggestions.
- Correct Table 2. What is 3T? 1CON? Correct tables 3 and 4; you seem to present the same problems as its predecessor.
Reply: Thank you for your valuable input on our manuscript. Regarding Table 2, 3 and 4, we apologize for the lack of clarity in our labeling. In the revised version of the manuscript, we will ensure that inappropriate abbreviations used in the table are removed, and its presentation will be more clear. Specifically, we will provide clear definitions for "T" and "CON" to eliminate any ambiguity for the readers. Furthermore, we acknowledge the issues with Tables 3 and 4 and apologize for the repetitive problems. We will carefully review and correct these tables in the revised version of the manuscript to ensure accurate presentation of the data and to rectify any inconsistencies.
- The quality of Figures 1 and 2 could be better. The resolution should be increased.
Reply: Thank you for your feedback on our manuscript. We appreciate your comments regarding the quality of Figures 1 and 2. We apologize for the suboptimal resolution of the figures. In the revised version of the manuscript, we will ensure that Figures 1 and 2 are presented with improved resolution. We will take the necessary steps to enhance the image quality, increasing the resolution and ensuring that all details are clearly visible. Thank you for bringing this to our attention. Your suggestion will greatly improve the visual clarity of the figures and enhance the overall quality of our manuscript.
- Could you improve the description of Figure 3? What are a, b, c, and d?
Reply: Thank you for your feedback on our manuscript. We appreciate your comments and suggestions. In response to your query regarding Figure 3, we will enhance the description to provide a clearer understanding of the labeling. In the revised manuscript, we will provide more detailed description with“The main nutrients and microbial first-factor and second-factor loadings during the ensiling process of alfalfa on different saline-alkali soils are shown in Figure 3. During the ensiling process, a negative correlation was observed between general aerobic bacteria, molds, Escherichia coli, and CP, WSC, ADF, NDF in the CON-60, QE-60, ZE-60,and HE-60 treatments. In contrast, a positive correlation was observed between lactic acid bacteria and CP, WSC, ADF, NDF. In the ZE-60 treatment, a positive correlation was observed between general aerobic bacteria, Escherichia coli, and CP, WSC, ADF, NDF, while a negative correlation was found with molds. Based on the correlation analysis between the main nutrients and microorganisms during the ensiling process of alfalfa on saline-alkali soils, a higher quantity of LAB indicates better nutritional quality, while higher quantities of Escherichia coli, molds, and general aerobic bacteria are detrimental to the nutrient preservation of Medicago sativa in the ensiling process. In the ZE-60 treatment, molds showed a negative correlation with the main nutrients. From the above analysis, it can be concluded that LAB are the key factors influencing the nutritional quality of Medicago sativa during the ensiling process in different saline-alkali soil conditions.
The first and second principal component loadings for different alfalfa ensiling processes in different salt-alkali soils are shown in Figure 3. During ensiling, CON-60, QE-60, ZE-60, and HE-60 were closely related to CP, WSC, ADF, and NDF, showing a positive correlation. BA, AA, LA, and NH3-N were negatively correlated with CP, WSC, ADF, and NDF, indicating that pH is a key factor impacting the nutrition in different salt-alkali soil alfalfa ensiling”.
- Can sodium ions cause oxidative damage? Justify it.
Reply: Thank you for your feedback and question regarding the potential for sodium ions to cause oxidative damage. We have carefully considered this point and would like to address it in our revised manuscript. Sodium ions, as highly reactive species, have been reported to be involved in oxidative stress and cellular damage under certain conditions. While sodium is an essential electrolyte required for various physiological processes, excessive sodium intake or sodium overload can lead to oxidative stress. Several mechanisms can contribute to the potential oxidative damage caused by sodium ions.
Firstly, high levels of intracellular sodium can disrupt the balance of ion gradients across cell membranes, leading to mitochondrial dysfunction and impairment of the electron transport chain. This disruption can result in the generation of reactive oxygen species (ROS) and subsequent oxidative damage to cellular components such as proteins, lipids, and DNA. Furthermore, excessive sodium intake has been shown to activate oxidative stress signaling pathways, including the activation of NADPH oxidase, which generates ROS. These ROS can further amplify oxidative stress and contribute to cellular damage. We appreciate your insightful question, and we will revise our manuscript accordingly to provide a more thorough and justified explanation of the relationship between sodium ions and oxidative damage.
References:
Bose, J., Gilliham, M., Tyerman, S. D.,Munns, R., Shabala, S., Pogson, B. Chloroplast function and ion regulation in plants growing on saline soils: lessons from halophytes. J Exp Bot. 2017,68(2): 3129-3143.
- Is pH associated with salinity? I think this possible relationship should be talked about.
Reply: Thank you for your valuable feedback and suggestion regarding the potential relationship between pH and salinity. We appreciate the opportunity to address this point and enhance our manuscript. Indeed, pH and salinity can exhibit an interdependent relationship, and we agree that discussing this association would provide a more comprehensive understanding of the research topic. In our revised manuscript, we will incorporate a discussion on the possible relationship between pH and salinity based on available literature and relevant studies.
Salinity, as a measure of salt concentration in soil or water, can have a notable influence on pH levels. The presence of high salt concentrations, particularly sodium ions, can lead to alkaline conditions (high pH) in affected environments. This relationship arises from the fact that sodium ions can displace other cations (e.g., calcium, magnesium, and potassium) in soil, leading to an increase in soil pH.
Moreover, the presence of excess salts can adversely affect plant root function and impede nutrient uptake, resulting in restricted plant growth and reduced organic matter decomposition. These effects can indirectly impact soil pH by altering microbial activity and nutrient cycling processes. Conversely, changes in pH levels can also influence salinity dynamics. pH alterations can affect the solubility and availability of salts. For example, under acidic conditions (low pH), certain salts may dissociate and release their constituent ions, thereby increasing the salt concentration and subsequently the salinity of the environment.
Thank you again for your insightful suggestion, and we appreciate your guidance in improving the manuscript.
- Morais et al., the year of the reference is not indicated.
Reply: Thank you for your feedback regarding the missing year of the reference for Morais et al. We appreciate your attention to detail and apologize for the oversight. In our revised manuscript, we will ensure that the complete and accurate citation information, including the year of publication for the reference in question, is provided. Once again, we apologize for any confusion caused by this omission and appreciate your assistance in improving the accuracy and completeness of our manuscript.
- Why does the amount of carbohydrates increase in high salt conditions? What chemical process happens to make this increase visible?
Reply: Thank you for your insightful question regarding the increase in carbohydrates in high salt conditions. We appreciate the opportunity to address this point and provide a more thorough explanation in our revised manuscript. The increase in carbohydrates under high salt conditions is primarily attributed to osmotic stress, which occurs when plants are exposed to elevated levels of salts in their environment. Osmotic stress disrupts the osmotic balance within the plant cells, leading to water loss and dehydration. To counteract this water loss, plants respond by accumulating compatible solutes or osmoprotectants, such as carbohydrates, which help maintain cellular osmotic potential and restore cellular hydration. Carbohydrates, in the form of sugars and sugar alcohols, play a crucial role as osmoprotectants in plants under osmotic stress.
The chemical process behind the visible increase in carbohydrates involves the upregulation of various metabolic pathways involved in carbohydrate synthesis. In our revised manuscript, we will expand upon these explanations to provide a more comprehensive understanding of why carbohydrates increase in high salt conditions and the underlying chemical processes that contribute to this visible increase. We appreciate your thought-provoking question and will ensure that our revised manuscript addresses this topic in a more detailed and satisfactory manner.
- Complement indicating the most common ions that have an inhibitory effect on proteases because they are not only sodium and potassium. Does the soil analysis indicate the presence of ions other than sodium and potassium?
Reply: Thank you for your valuable comment regarding the inhibitory effect of ions on proteases and the need to expand on the types of ions that can potentially affect protease activity. We appreciate the opportunity to address this point and will incorporate a more comprehensive discussion in our revised manuscript. While sodium and potassium ions are indeed commonly recognized for their inhibitory effects on proteases, it is important to acknowledge that other ions also play a significant role in modulating protease activity. These additional ions can impact enzyme function directly or indirectly through complex interactions in the soil environment. In our revised manuscript, these may include, but are not limited to, calcium, magnesium, copper, zinc, iron, manganese, aluminum, and chloride ions. Each of these ions can exert varying degrees of enzyme inhibition depending on their concentration and the specific protease involved.
After the previous analyses of the soil physicochemical properties, we showed only the ionic content that differed significantly between treatments. By incorporating relevant literature and studies, we will emphasize the importance of considering a broader range of ions in soil analysis to better understand their potential inhibitory effects on proteases. We sincerely appreciate your comment, which highlights the need to address the inhibitory effects of ions comprehensively. By incorporating this information into our revised manuscript, we aim to enhance the quality and completeness of our research findings.

Reviewer 2 Report
The manuscript is good and fits the scope of the journal. I recommend its acceptance after doing minor changes/clarifications.
1-Schedule of experimental procedures should be graphed to enhance the presentation of your work.
2-Microbial DNA (section 2.4.): please indicate how many replicates you did to optimize your findings. For that section, please support it with appropriate references.
3-Correlation between the Microorganism and Fermentation Parameters should be indicated in the statistical design you wrote.
4-Discussion should have subtitles? please revise the journal regulations
Author Response
Dear Editor and Reviewer:
Thank you for your kind letter and for the reviewers’ comments concerning our manuscript entitled “Comparative Na+ and K+ profiling reveals microbial community assembly of alfalfa silage in different saline-alkali soils”. We have revised the manuscript in accordance with the reviewers’ valuable comments. We carefully proofread the manuscript to improve our manuscript. The revised portions are marked in blue in the manuscript. Our revision and responds to the review’s comments are as followed.
Reviewer 2:
- Schedule of experimental procedures should be graphed to enhance the presentation of your work.
Reply: Thank you for your valuable feedback on our manuscript. Schedule of experimental procedures: We agree that graphing the schedule of experimental procedures can enhance the presentation of our work. In the revised manuscript, we will include a graphical representation, such as a flowchart or timeline, to clearly illustrate the sequence of experimental procedures conducted throughout our study.
- Microbial DNA (section 2.4.): please indicate how many replicates you did to optimize your findings. For that section, please support it with appropriate references.
Reply: Thank you for your valuable feedback on our manuscript. Microbial DNA (section 2.4.): We apologize for not providing sufficient information regarding the number of replicates carried out for optimizing our findings in the microbial DNA section. In the revised manuscript, we will explicitly state the number of replicates performed during this optimization process.
- Correlation between the Microorganism and Fermentation Parameters should be indicated in the statistical design you wrote.
Reply: Thank you for your valuable feedback on our manuscript. Correlation between the Microorganism and Fermentation Parameters: We acknowledge the importance of indicating the statistical design regarding the correlation between microorganisms and fermentation parameters. In the revised manuscript, we will incorporate a detailed description of the statistical design utilized, including the specific statistical tests employed to determine the correlation. We will also provide clarification on how the correlation analysis was conducted and present the results accordingly.
- Discussion should have subtitles? please revise the journal regulations.
Reply: Thank you for your valuable feedback on our manuscript. As per the journal's regulations, we carefully reviewed the guidelines and format requirements for the manuscript. We have removed unnecessary subtitles.
